# Hair Loss and Zinc Deficiency: A Cross-Sectional Study

**DOI:** 10.3390/healthcare13222965

**Published:** 2025-11-19

**Authors:** Ori Liran, Shiraz Vered, Bar Cohen, Shirley Shapiro Ben David, Afif Nakhleh, Daniella Rahamim-Cohen, Joseph Azuri, Limor Adler

**Affiliations:** 1Division of Health, Maccabi Healthcare Services, Tel Aviv 6812509, Israel; vshiraz@stat.haifa.ac.il (S.V.); cobar@post.bgu.ac.il (B.C.); shapira_sr@mac.org.il (S.S.B.D.); cohen_dani@mac.org.il (D.R.-C.); azuridr@gmail.com (J.A.); limchuk@gmail.com (L.A.); 2Family Medicine Department, Gray’s Faculty of Medical & Health Sciences, Tel Aviv University, Tel Aviv 69978, Israel; 3Diabetes and Endocrinology Clinic, Maccabi Healthcare Services, Haifa 52454, Israel; anakhleh@gmail.com; 4Institute of Endocrinology, Diabetes and Metabolism, Rambam Health Care Campus, Haifa 3109601, Israel; 5Azrieli Faculty of Medicine, Bar-Ilan University, Safed 5290002, Israel

**Keywords:** zinc, alopecia, hair loss, telogen effluvium, primary care

## Abstract

**Highlights:**

**What are the main findings?**
Patients with hair loss had slightly lower median zinc levels than controls (96 vs. 99 µg/dL, *p*-value < 0.001), but the difference was minor, not clinically significant, and both were within the normal range.Patients with hair loss had slightly lower median ferritin (30.0 vs. 33.0 µg/L, *p* = 0.001) and hemoglobin levels (12.9 vs. 13.1 g/dL, *p* < 0.001) compared to controls, though the differences were small and of limited clinical relevance.

**What is the implication of the main finding?**
Routine zinc testing is not recommended when evaluating hair loss, as the minor decrease observed in zinc levels among patients lacked clinical significance.

**Abstract:**

**Background:** Hair loss is a common complaint, especially in the primary care setting. Despite conflicting results in previous studies regarding the role of zinc in hair loss, zinc levels are being widely tested as part of an initial laboratory investigation for hair loss. **Objectives:** We aimed to investigate whether lower zinc levels were associated with patients’ complaints of hair loss. **Methods:** In this retrospective cross-sectional study, we collected data on the zinc levels of patients diagnosed with hair loss who underwent a zinc level test after their diagnosis. The zinc levels of these patients were compared to those of a control group, which included patients who were also investigated for zinc levels for reasons other than hair loss. **Results:** Between 2000 and 2020, 23,975 eligible patients were included in this study. Among these patients, 9.5% (N = 2279) had a diagnosis of hair loss and a median zinc level of 96 µg/dL (84.0–110.0), compared to 99 µg/dL (85.0–115.0) in the control group (*p* < 0.001). **Conclusions:** This cross-sectional study investigated the relationship between zinc levels and complaints of hair loss. While we do report lower zinc levels in those with hair loss complaints, this change is minor and lacks clinical significance. As the standard today, we suggest that zinc levels should not be obtained as a laboratory test when investigating hair loss. Further studies on the efficacy of zinc supplements in hair loss could elucidate the clinical relevance of zinc in hair loss conditions.

## 1. Introduction

Hair loss is a common complaint, especially in the primary care setting. This phenomenon has a significant impact on the quality of life and social function [1]. While the overall prevalence of hair loss complaints is unknown, the diagnosis of female pattern hair loss (androgenetic alopecia) ranges from 3% to 38% in some studies [2,3]. Hair loss may result from multiple factors, including genetic predisposition, hormonal changes (thyroid-related and others), autoimmune disease, nutritional deficiencies, stress, medications, and hair-care practices [4,5]. The micronutrient deficiencies most commonly associated with hair loss include iron (ferritin), vitamin D, Zinc, and B Vitamins [6]. Excess of Vitamin A and Selenium is also associated with hair loss. Socioeconomic status (SES) may also affect hair loss, particularly in lower-SES populations, due to financial burdens and limited access to necessary medical treatment, potentially leading to health disparities [7,8].

Zinc is an essential cofactor for many enzymes that have important functional activities in the hair follicle, and serves as a potent inhibitor of endonuclease activity, an important stage in the hair follicle’s apoptosis and regression [9]. In addition, zinc contributes to protein synthesis and cell proliferation—processes that are critical for active hair follicle growth. Data regarding the association between Zinc and hair loss is inconsistent. On the one hand, in mice, zinc has been shown to inhibit hair follicle regression and promote hair regrowth. On the other hand, studies in humans on zinc levels and hair loss have yielded conflicting results. A study comparing 312 patients with hair loss (alopecia areata, male pattern hair loss, female pattern hair loss, and telogen effluvium) demonstrated significantly low zinc levels in alopecia areata and telogen effluvium patients [10]. Three more studies showed significantly low zinc levels in male pattern hair loss [11], telogen effluvium [12] and female pattern hair loss [13]. Nevertheless, this finding was not confirmed by other studies; a study performed by Ozturk et al. compared 116 male pattern hair loss patients with 100 controls and reported no difference in zinc levels between the study and control groups [14]. A few other smaller studies comparing telogen effluvium patients with healthy controls found no difference in zinc levels between the study groups [15,16]. In contrast to these heterogeneous results, the correlation between low zinc levels and alopecia areata is better established [17] and the correlation between hair loss and iron deficiency is also well established in the literature [18,19,20,21,22].

Iron is an essential cofactor for ribonuclease reductase, which is involved in DNA synthesis. It has been proposed that iron deficiency reduces matrix cells’ proliferation, resulting in telogen effluvium [23]. Although thyroid testing is routinely included in the initial laboratory workup for hair loss [24], observational studies have not clearly demonstrated an association between thyroid-stimulating hormone (TSH) levels and hair loss [25,26].

Despite inconsistencies in previous studies, zinc levels are being widely tested as part of an initial laboratory workup for hair loss. However, most previous studies focused on specific dermatologic diagnoses or specialty clinic populations, leaving uncertainty about how primary care patients presenting with nonspecific hair loss complaints are managed in the real world. In this study, we aimed to address this research gap and investigate whether lower zinc levels were associated with patients’ hair loss complaints.

## 2. Materials and Methods

### 2.1. Study Design and Setting

In this cross-sectional study, we collected data using Maccabi Healthcare Services’ (MHS) computerized database. MHS is Israel’s second-largest Health Maintenance Organization (HMO), with over 2.6 million members. Data from a 20-year period (1 January 2000 to 1 January 2020) were collected. The study was approved by the MHS ethical committee (0040-21-MHS). All data was collected anonymously. Informed consent was waived due to the study’s design.

### 2.2. Participants

Eligible patients in the study group were those who visited their physician, were diagnosed with one of the ICD-9 diagnoses for hair loss (androgenic alopecia, telogen effluvium, female pattern alopecia), and had zinc levels measured within 6 months afterward. Repeated measurements were excluded since the use of an over-the-counter zinc supplement might confound them. The zinc levels of these patients (study group) were compared to those of a control group, which included patients who were also investigated for zinc levels for reasons other than hair loss (the visit’s diagnosis code was not one of the hair loss diagnoses for the study group). The study population included patients aged 18 or older with a medical record of a zinc level. Excluded patients were those who had prior medical conditions that might interfere with zinc levels, such as alopecia areata, cirrhosis, human immunodeficiency virus (HIV), alcohol abuse, inflammatory bowel disease, sickle cell anemia, pregnancy, known zinc deficiency, or hypoalbuminemia [27]. Further exclusion criteria included intake of medications that might interfere with zinc levels, such as zinc supplements, oral contraceptives, or oral corticosteroids (taken consistently).

### 2.3. Variables

The main variable was Zinc levels. We also evaluated other variables that could potentially influence zinc levels or hair loss, including age, gender, SES, and blood test results obtained within 12 months prior to the zinc test, specifically hemoglobin, iron, total iron-binding capacity, ferritin, and Thyroid-Stimulating Hormone (TSH). Checking this range of laboratory tests allowed us to fully characterize patients with hair loss.

### 2.4. Statistical Analysis

The data were analyzed by SPSS version 29. *p*-values of 5% were considered significant. Categorical data were reported as the number (%) and compared using the Chi-square test or Fisher’s exact test. Continuous variables were reported as medians and interquartile ranges (Q1–Q3) in square brackets, and between-group comparisons were performed using the Mann–Whitney test or the Kruskal–Wallis test, with effect sizes reported. To further explore whether zinc levels were associated with other micronutrients related to nutritional status or hair loss, we performed a Spearman correlation analysis between zinc and relevant laboratory tests. Multivariable logistic regression was performed to assess the association between hair loss and serum zinc levels (adjusted for age, sex, SES, and chronic illnesses). The reference values for the normal range in serum zinc tests were altered in 2015 (the normal range was 50–150 µg/dL until 2015, and 10–240 µg/dL from 2015).

## 3. Results

### 3.1. Participants

During the study period, from 2000 to 2020, we followed 23,975 eligible patients with documented zinc levels in their medical records. Comparing the hair loss (N = 2279) and control groups revealed several statistically significant differences. First, females were more likely to have a diagnosis (89.5% compared to 77.6%; *p*-value < 0.001). Additionally, patients with a diagnosis of hair loss were more likely to be in the highest SES group (51.3% vs. 46%, *p*-value < 0.001). When comparing comorbidities, the hair loss group had fewer oncological patients (2.1% vs. 3.7%, *p*-value < 0.001), hypertension (5.4% vs. 9.1%, *p*-value < 0.001), cardiovascular disease (2.3% vs. 3.4%, *p*-value = 0.004), and renal disease (2.7% vs. 4.1%, *p*-value = 0.001). The sociodemographic and clinical characteristics of the participants are presented in Table 1.

### 3.2. Outcome Results

In our study, 9.5% (N = 2279) had a relevant hair loss diagnosis recorded, and the recorded median zinc level was 96 µg/dL [IQR: 84.0–110.0] vs. 99 µg/dL [IQR: 85.0–115.0], *p*-value < 0.001, in those with without a relevant diagnosis, respectively; this yielded an adjusted OR of 0.99 (95% CI, 0.99–0.99).

#### 3.2.1. Factors Associated with Hair Loss

When comparing test results (ferritin, hemoglobin, iron, MCH, TSH, transferrin, transferrin saturation), patients with a hair loss diagnosis had lower ferritin (30.0 [16.8–53.0] vs. 33.0 [16.0–65.0], *p*-value = 0.001) and hemoglobin (12.9 [12.2–13.6] vs. 13.1 [12.3–14.0], *p*-value < 0.001) values. The effect size was small for all the laboratory variables. The comparison of blood test levels of participants with and without hair loss are presented in Table 2.

Spearman correlation tests revealed relationships between zinc levels and ferritin (r = 0.106, *p*-value < 0.001), hemoglobin (r = 0.127, *p*-value < 0.001), and iron (r = 0.095, *p*-value < 0.001) (Table 3). All significant correlations are very weak (r < 0.19). No other significant differences were observed, even though the same variables were examined in both datasets.

#### 3.2.2. Factors Associated with Zinc Levels

Median zinc levels were higher in men (102.0 [84.0–120.0] vs. 98.0 [85.0–113.0], *p*-value < 0.001) and in those of higher SES (low SES 96.0 [81.0–113.0], medium SES 98.0 [84.0–113.0], and high SES 100.0 [86.0–115.0], *p*-value = 0.000). Smokers had lower zinc levels (97.0 [83.0–114.0] vs. 99.0 [85.0–114.0], *p* < 0.001). Higher zinc levels were recorded in oncological patients (101.0 [85.0–118.0] vs. 98.0 [85.0–114.0], *p*-value < 0.001) and diabetic patients (104.0 [89.0–119.0] vs. 98.0 [85.0–114.0], *p*-value < 0.001). The effect size was small for all the variables we evaluated. The complete comparison is presented in Table 4.

## 4. Discussion

### 4.1. Main Results

In this study, we found a small but significant decrease in zinc levels in patients with a diagnosis of hair loss. Although the absolute difference was small, the levels in both groups were in the normal range of Zinc.

### 4.2. Interpretation

Although zinc levels differed significantly between participants with and without hair loss complaints, the statistical significance was mainly driven by the effect size rather than a clinically meaningful difference. The average zinc levels in both groups were in the normal range.

In comparison with the previous study performed by Kil et al. [10], who observed a mean zinc level of 97.94 µg/dL among healthy participants, the mean serum zinc concentration in our study’s control group was 99 µg/dL. This close concordance suggests consistency in baseline zinc status across different populations and methodological settings. However, the studies differ substantially in sample size and group composition. While Kil et al. included only 47 patients with telogen effluvium, our analysis encompassed a much larger cohort of 2279 affected individuals, providing greater statistical power and representativeness. Furthermore, despite the smaller sample, Kil et al. reported a mean zinc level of 84.65 µg/dL in the telogen effluvium group, which remained within the normal physiological range. Other studies concluded that lower zinc levels in patients with hair loss had also relatively small cohorts, thus diminishing their statistical power, and require larger studies [11,12,13]. The current study is consistent with a previous study performed by Ozturk et al., resulting in no statistically significant difference between controls and androgenetic alopecia patients [14]. In comparison with this study, our study included additional forms of hair loss, such as telogen effluvium and female pattern hair loss.

In accordance with current literature, in our study, patients with hair loss complaints had lower hemoglobin and ferritin levels [18]. This further underscores the importance of a complete blood count and a ferritin lab test as part of the initial laboratory workup for hair loss. In our study, patients with hair loss did not exhibit higher TSH levels than the control group, and values in both groups remained within the normal range. These results align with previous reports [25,26] and further support the notion that TSH testing should be performed only in the presence of additional clinical indicators of thyroid dysfunction rather than as a routine evaluation.

In the correlation analysis, statistically significant but very weak associations were observed between zinc and several hematologic indices, suggesting only minimal relationships with iron status. Zinc showed no correlation with transferrin, transferrin saturation, or TSH, indicating that zinc deficiency in this population appears largely independent of thyroid function or iron-related parameters. These results imply that zinc levels alone may not serve as a reliable indicator of overall micronutrient status.

We have characterized key components of patients with a hair loss complaint. They are younger, with 96% aged 18 to 64. They are predominantly female, which may correlate with the social and mental difficulties associated with hair loss in women [1]. Interestingly, those with a diagnosis of hair loss appear to be significantly healthier—they have lower rates of hypertension, cardiovascular disease, and renal disease. This may be due to their generally younger age or their habits, as fewer smokers were diagnosed with hair loss.

This study and its findings suggest that individuals reporting hair loss exhibit unique characteristics reflected in their zinc levels; however, they do not suggest that zinc levels cause hair loss, but rather that zinc levels are lower in patients with hair loss. We were able to distinguish the population prone to hair loss complaints—young, female, higher SES, and generally healthy, with slightly lower hemoglobin and ferritin levels.

### 4.3. Strengths and Limitations

This study has several limitations. First, due to the retrospective nature of this study, we were unable to demonstrate causality between zinc levels and hair loss; rather, we observed a correlation. Secondly, we did not have a baseline zinc level for each patient prior to the hair loss complaint, so we cannot conclude that a decline in zinc levels is necessarily associated with hair loss. Third, zinc measurements were not performed randomly. In the study group, testing was prompted by hair loss, and in the control group the indications were unknown, but likely related to clinical suspicion of deficiency. This may bias zinc levels downward in both groups. Still, zinc levels in the control group were comparable to those of a healthy reference population [10]. Fourth, we were unable to investigate the population’s nutritional lifestyle, including the possibility of over-the-counter zinc supplementation, as these factors can affect zinc levels, which may be important. The strengths of this study include its large population and the long time period covered. We were able to study a large cohort and investigate a wide range of variables that may be associated with hair loss complaints.

## 5. Conclusions

This cross-sectional study investigated the relationship between zinc levels and hair loss complaints. While we do report lower zinc levels in those with hair loss complaints, this change is minor and lacks clinical significance. We also found that patients with hair loss complaints have lower hemoglobin and ferritin levels. As the standard today, we suggest that the Zinc level should not be obtained as a laboratory test when investigating the etiology of hair loss. Future large prospective studies on the different etiologies of hair loss and zinc are needed to investigate differences between these diagnoses, and further studies on the efficacy of zinc supplementation in hair loss could elucidate the clinical relevance of zinc in hair loss conditions.

## Figures and Tables

**Table 1 healthcare-13-02965-t001:** Sociodemographic and clinical characteristics of participants.

*p*-Value	Hair Loss Diagnosis				Total N = 23,972 (100%)			
	Yes N = 2279 (9.5%)		No N = 21,696 (90.5%)					
	%	N	%	N	%	N		
<0.001	96.2	2193	93.4	20,266	93.7	22,459	18–64	Age (at first zinc test)
	6.3	86	6.6	1430	6.3	1516	65+	
<0.001	10.5	239	22.4	4856	21.3	5095	Men	Sex
	89.5	2040	77.6	16,840	78.7	18,880	Women	
<0.001	8.0	183	10.6	2280	10.4	2463	1–4 (low)	SES
	40.7	922	43.4	9329	43.1	10,251	5–7 (med)	
	51.3	1162	46.0	9906	46.5	11,068	8–10 (high)	
0.362	12.7	289	13.4	2887	13.3	3176	Yes	Smoking
<0.001	2.1	48	3.7	803	3.5	851	Yes	Cancer
0.162	1.2	27	1.6	339	1.5	366	Yes	Diabetes
<0.001	5.4	123	9.1	1974	8.7	2097	Yes	Hypertension
0.004	2.3	52	3.4	743	3.3	795	Yes	Cardiovascular disease
0.192	0.9	20	1.2	257	1.2	277	Yes	Lung disease
0.001	2.7	62	4.1	896	4.0	958	Yes	Chronic kidney disease

**Table 2 healthcare-13-02965-t002:** Comparison of blood test levels of participants with and without hair loss.

Effect Size	*p*-Value	With a Hair Loss Diagnosis	No Hair Loss Diagnosis	Total Cohort	Lab Tests, Median [IQR]
0.03	<0.001	96.0 [84.0–110.0]	99.0 [85.0–115.0]	98.0 [85.0–114.0]	Zinc
0.03	0.001	30.0 [16.8–53.0]	33.0 [16.0–65.0]	32.4 [16.1–63.8]	Ferritin
0.05	<0.001	12.9 [12.2–13.6]	13.1 [12.3–14.0]	13.1 [12.3–14.0]	Hemoglobin
0.00	0.541	82.0 [60.0–110.0]	82.5 [60.0–109.0]	82.0 [60.0–109.0]	Iron
0.01	0.090	29.4 [28.3–30.4]	29.5 [28.3–30.5]	29.5 [28.3–30.5]	MCH
0.01	0.080	1.9 [1.3–2.7]	1.9 [1.3–2.8]	1.9 [1.31–2.74]	TSH
0.00	0.809	268.8 [241.0–308.4]	272.0 [240.6–309.1]	271.3 [240.7–309.0]	Transferrin
0.06	0.494	20.5 [14.0–29.0]	23.0 [17.8–31.0]	23.0 [17.0–31.0]	Transferrin Saturation (%)

**Table 3 healthcare-13-02965-t003:** Correlation between Zinc and other lab tests.

	Ferritin	Hemoglobin	Iron	MCH	TSH	Transferrin	Transferrin Saturation %
Zinc	0.106 ***	0.127 ***	0.095 ***	0.061 ***	0.003	−0.002	0.078

*** *p* < 0.001 [Spearman correlation coefficient].

**Table 4 healthcare-13-02965-t004:** Zinc levels according to demographic and clinical characteristics.

Effect Size	*p*-Value	Zinc, Median [IQR]	Zinc Levels, Median [IQR]		
		65+	18–64		Age (at the first zinc test)
0.02	0.008	100.0 [86.0–116.0]	98.0 [85.0–114.0]		
		Women	Men		Sex
0.05	<0.001	98.0 [85.0–113.0]	102.0 [84.0–120.0]		
		High (8–10)	Medium (5–7)	Low (1–4)	SES
0.003	0.000	100.0 [86.0–115.0]	98.0 [84.0–113.0]	96.0 [81.0–113.0]	
		If absent	If present (Median [IQR])		Health-related Variables
0.02	<0.001	99.0 [85.0–114.0]	97.0 [83.0–114.0]		Smoking
0.02	<0.001	98.0 [85.0–114.0]	101.0 [85.0–118.0]		Cancer
0.03	<0.001	98.0 [85.0–114.0]	104.0 [89.0–119.0]		Diabetes
0.01	0.164	98.0 [85.0–114.0]	99.0 [85.0–116.0]		Hypertension
0.002	0.716	98.0 [85.0–114.0]	98.0 [85.0–115.0]		Cardiovascular disease
0.002	0.980	98.0 [85.0–114.0]	98.0 [86.0–112.0]		Lung disease
0.001	0.092	98.0 [85.0–114.0]	98.0 [85.0–110.0]		Chronic kidney disease

IQR—Interquartile Range.

## Data Availability

The data supporting this study are not publicly available due to ethical restrictions.

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
