# Peer review of "Hair Loss and Zinc Deficiency: A Cross-Sectional Study"

_healthcare, 2025, doi:10.3390/healthcare13222965_

Round 1
Reviewer 1 Report
Comments and Suggestions for Authors
- Introduction
- The background information on the role of zinc in hair loss and its biological relevance should be further elaborated in the Introduction. Additionally, the authors should clearly highlight the novelty and research gap that this study aims to address.
- It would be beneficial to include previous studies or references discussing the various factors investigated in this work that are related to hair loss, to provide stronger scientific context and support for the study rationale.
- Study Design
- Please clarify which type of hair loss was targeted in this study. Was it necessary to categorize different types of alopecia to ensure specificity of the findings? Also, please explain why Alopecia Areata was excluded from this research and whether its exclusion affects the generalizability of the results.
- The authors should justify the inclusion of socioeconomic status (SES) as a study variable and describe its potential relationship with hair loss or associated biological factors.
- Please explain the rationale for selecting the specific laboratory markers analyzed in this study. Why were these particular markers chosen, and how are they relevant to hair loss mechanisms?
- Discussion
- The correlation analysis should be discussed in greater detail. What was the purpose of performing this analysis, and what do the results reveal about the relationships among the studied variables?
- The laboratory markers measured should be further discussed in relation to their biological or clinical relevanceto hair loss. Please include supporting evidence or references to strengthen these interpretations.
- The authors should compare their findings with previous studies, highlighting whether the present results are consistent with or differ from prior research, and discussing possible explanations for these similarities or discrepancies.
- The practical significance of this research should be clearly stated. How can the findings be applied in clinical or medical practice, and what potential implications do they hold for future studies or therapeutic approaches?
Author Response
Reviewer 1
- Introduction
- The background information on the role of zinc in hair loss and its biological relevance should be further elaborated in the Introduction. Additionally, the authors should clearly highlight the novelty and research gap that this study aims to address.
ANSWER:
We added the following section to the introduction:
Zinc is an essential cofactor for many enzymes that have important functional activities in the hair follicle, and serves as a potent inhibitor of endonuclease activity, an important stage in the hair follicle's apoptosis and regression 9. In addition, zinc contributes to protein synthesis and cell proliferation—processes that are critical for active hair follicle growth. Data regarding the association between Zinc and hair loss is inconsistent. On the one hand, in mice, zinc has been shown to inhibit hair follicle regression and promote hair regrowth. On the other hand, studies in humans on zinc levels and hair loss have yielded conflicting results. (Introduction, p2, 56-63)
We also added:
Despite inconsistencies in previous studies, zinc levels are being widely tested as part of an initial laboratory workup for hair loss. However, most previous studies focused on specific dermatologic diagnoses or specialty clinic populations, leaving uncertainty about how primary care patients presenting with nonspecific hair loss complaints are managed in the real world. In this study, we aimed to address this research gap and investigate whether lower zinc levels were associated with patients' hair loss complaints. (Introduction, p2, 80-85)
- It would be beneficial to include previous studiesor references discussing the various factors investigated in this work that are related to hair loss, to provide stronger scientific context and support for the study rationale.
ANSWER:
Regarding the various factors, including later in the study, we've added this section to the introduction section:
Hair loss may result from multiple factors, including genetic predisposition, hormonal changes (thyroid-related and others), autoimmune disease, nutritional deficiencies, stress, medications, and hair-care practices 4,5. The micronutrient deficiencies most commonly associated with hair loss include iron (ferritin), vitamin D, Zinc, and B Vitamins 6. Excess of Vitamin A and Selenium is also associated with hair loss. (Introduction, p2, lines 48-53)
- Study Design
- Please clarify which type of hair losswas targeted in this study. Was it necessary to categorize different types of alopecia to ensure specificity of the findings? Also, please explain why Alopecia Areata was excluded from this research and whether its exclusion affects the generalizability of the results.
ANSWER: - The types of hair loss targeted in this study are elaborated in 'materials and methods' in the 'participants section'.
Eligible patients in the study group were those who visited their physician, were diagnosed with one of the ICD-9 diagnoses for hair loss (androgenic alopecia, telogen effluvium, female pattern alopecia), and had zinc levels taken afterward, within 6 months. (Participants, p3, 96-98)
The reason Alopecia Areata was excluded from this study, as mentioned in the 'participants' section, is that the association between zinc deficiency and Alopecia Areata is more established in current literature. This is mentioned in the Introduction section. In contrast to these heterogeneous results, the correlation between low zinc levels and alopecia areata is better established 17 (Introduction, p2, 71-73)
- The authors should justify the inclusion of socioeconomic status (SES) as a study variable and describe its potential relationship with hair loss or associated biological factors.
ANSWER
We added this to the introduction
Socioeconomic status (SES) may also affect hair loss, particularly in lower-SES populations, due to financial burdens and limited access to necessary medical treatment, potentially leading to health disparities 7,8. (Introduction, P2 53-55)
- Please explain the rationalefor selecting the specific laboratory markers analyzed in this study. Why were these particular markers chosen, and how are they relevant to hair loss mechanisms?
ANSWER
We added this to the introduction.
Hair loss may result from multiple factors, including genetic predisposition, hormonal changes (thyroid-related and others), autoimmune disease, nutritional deficiencies, stress, medications, and hair-care practices 4,5. The micronutrient deficiencies most commonly associated with hair loss include iron (ferritin), vitamin D, Zinc, and B Vitamins 6. Excess of Vitamin A and Selenium is also associated with hair loss. (Introduction, P2, lines 48-53)
- Discussion
- The correlation analysisshould be discussed in greater detail. What was the purpose of performing this analysis, and what do the results reveal about the relationships among the studied variables?
ANSWER:
We added this to the statistical analysis:
To further explore whether zinc levels were associated with other micronutrients related to nutritional status or hair loss, we performed a Spearman correlation analysis between zinc and relevant laboratory tests. (Materials and Methods, p3, lines 122-4)
We also added this to the discussion:
In the correlation analysis, statistically significant but very weak associations were observed between zinc and several hematologic indices, suggesting only minimal relationships with iron status. Zinc showed no correlation with transferrin, transferrin saturation, or TSH, indicating that zinc deficiency in this population appears largely independent of thyroid function or iron-related parameters. These results imply that zinc levels alone may not serve as a reliable indicator of overall micronutrient status. (discussion, p6, lines 216-222)
- The laboratory markersmeasured should be further discussed in relation to their biological or clinical relevance to hair loss. Please include supporting evidence or references to strengthen these interpretations.
ANSWER:
We discussed the results regarding iron and hemoglobin in the discussion part –
In accordance with current literature, in our study, patients with hair loss complaints have lower hemoglobin and ferritin levels 18. This further underscores the importance of a complete blood count and a ferritin lab test as part of the initial laboratory workup for hair loss. (discussion, p6, 208-211)
Furthermore, we have added another section to the introduction part – regarding the laboratory markers: Iron is an essential cofactor for ribonuclease reductase, which is involved in DNA synthesis. It has been proposed that iron deficiency reduces matrix cells' proliferation, resulting in telogen effluvium 23. Although thyroid testing is routinely included in the initial laboratory workup for hair loss24, observational studies have not clearly demonstrated an association between thyroid-stimulating hormone (TSH) levels and hair loss 25,26. (Introduction, p2, 75-79).
In addition, according to your comment, we have added this section to the discussion:
In our study, patients with hair loss did not exhibit higher TSH levels than the control group, and values in both groups remained within the normal range. These results align with previous reports 25,26 and further support the notion that TSH testing should be performed only in the presence of additional clinical indicators of thyroid dysfunction rather than as a routine evaluation. (Discussion, p6, 211-215)
- The authors should compare their findings with previous studies, highlighting whether the present results are consistent with or differ from prior research, and discussing possible explanations for these similarities or discrepancies.
ANSWER
We added this paragraph to the discussion section:
In comparison with the previous study performed by Kil et al. 10 who observed a mean zinc level of 97.94 µg/dl among healthy participants, the mean serum zinc concentration in our study's control group was 99 µg/dl. This close concordance suggests consistency in baseline zinc status across different populations and methodological settings. However, the studies differ substantially in sample size and group composition. While Kil et al. included only 47 patients with telogen effluvium, our analysis encompassed a much larger cohort of 2,279 affected individuals, providing greater statistical power and representativeness. Furthermore, despite the smaller sample, Kil et al. reported a mean zinc level of 84.65 µg/dL in the telogen effluvium group, which remained within the normal physiological range. Other studies concluded that lower zinc levels in patients with hair loss had also relatively small cohorts, thus diminishing their statistical power, and require larger studies 11–13. The current study is consistent with a previous study performed by Ozturk et al., resulting in no statistically significant difference between controls and androgenetic alopecia patients 14. In comparison with this study, our study included additional forms of hair loss, such as telogen effluvium and female pattern hair loss. (Discussion, P6, 193-207)
- The practical significanceof this research should be clearly stated. How can the findings be applied in clinical or medical practice, and what potential implications do they hold for future studies or therapeutic approaches?
ANSWER
We addressed these issues in the conclusion paragraph in the discussion section:
As the standard today, we suggest that the Zinc level should not be obtained as a laboratory test when investigating the etiology of hair loss. Future large prospective studies on the different etiologies of hair loss and zinc are needed to investigate differences between these diagnoses, and further studies on the efficacy of zinc supplementation in hair loss could elucidate the clinical relevance of zinc in hair loss conditions. (Discussion, p7, 253-258)

Reviewer 2 Report
Comments and Suggestions for Authors
This is a well-written and informative paper. A particular strength of the work is that, despite the finding of significantly lower zinc levels in the hair loss group, the investigators correctly concluded that the mean values both fell within the normal reference range and the magnitude of the differences was too small to be clinically relevant.
The only change I might suggest would be the addition of data that strengthens the rationale for this work investigating serum zinc levels in patients with hair loss. Specifically, the introduction to this paper would be strengthened by briefly citing previously published work documenting hair loss in patients with frank zinc deficiency, as opposed to only reporting work that measured zinc in patients with hair‐loss.
Author Response
Reviewer 2
This is a well-written and informative paper. A particular strength of the work is that, despite the finding of significantly lower zinc levels in the hair loss group, the investigators correctly concluded that the mean values both fell within the normal reference range and the magnitude of the differences was too small to be clinically relevant.
The only change I might suggest would be the addition of data that strengthens the rationale for this work investigating serum zinc levels in patients with hair loss. Specifically, the introduction to this paper would be strengthened by briefly citing previously published work documenting hair loss in patients with frank zinc deficiency, as opposed to only reporting work that measured zinc in patients with hair‐loss.
ANSWER
- Thank you very much for your elaborate comment. We have added the following to the introduction, regarding Zinc's biological role in hair regression. We have added this section to the introduction:
Zinc is an essential cofactor for many enzymes that have important functional activities in the hair follicle, and serves as a potent inhibitor of endonuclease activity, an important stage in the hair follicle's apoptosis and regression 9. In addition, zinc contributes to protein synthesis and cell proliferation—processes that are critical for active hair follicle growth. Data regarding the association between Zinc and hair loss is inconsistent. On the one hand, in mice, zinc has been shown to inhibit hair follicle regression and promote hair regrowth. (Introduction, P2, 56-62)

Reviewer 3 Report
Comments and Suggestions for Authors
The manuscript addresses the clinical relevance of serum zinc testing in patients presenting with hair loss. While the topic is important and the study is based on an impressively large database, the current version of the paper has significant methodological limitations that weaken the validity of its conclusions. Substantial revision is required before the manuscript can be considered for publication.
- Control Group Bias
The control group consists of patients who also underwent zinc testing, albeit for other reasons. This is not an adequate representation of a “healthy” or general population. It is highly likely that these patients have underlying clinical concerns that prompted zinc testing, introducing strong selection bias. A more appropriate control group is needed, or at minimum, the authors must acknowledge this as a major limitation and adjust their conclusions accordingly.
2. Heterogeneity of Hair Loss Diagnoses
The authors pool together multiple etiologies of hair loss (e.g., androgenetic alopecia, telogen effluvium, female pattern hair loss) despite the known differences in their pathophysiology and metabolic profiles. Without subgroup analysis, the findings are overly generalized and potentially misleading. At minimum, results should be stratified by diagnosis.
3. With a sample size of nearly 24,000 patients, statistically significant p-values are expected even for clinically trivial differences. The authors should report effect sizes (e.g., Cohen’s d) and address the magnitude, not merely the presence, of statistical differences. The current interpretation risks a “p-value fallacy.”
4. Key lifestyle and nutritional factors (diet, supplementation, vegetarianism, alcohol intake) are not addressed, despite being directly linked to zinc levels. This omission limits the strength of any causal inference or even correlation.
Author Response
Reviewer 3
The manuscript addresses the clinical relevance of serum zinc testing in patients presenting with hair loss. While the topic is important and the study is based on an impressively large database, the current version of the paper has significant methodological limitations that weaken the validity of its conclusions. Substantial revision is required before the manuscript can be considered for publication.
- Control Group Bias
The control group consists of patients who also underwent zinc testing, albeit for other reasons. This is not an adequate representation of a “healthy” or general population. It is highly likely that these patients have underlying clinical concerns that prompted zinc testing, introducing strong selection bias. A more appropriate control group is needed, or at a minimum, the authors must acknowledge this as a major limitation and adjust their conclusions accordingly.
ANSWER
Indeed, given the nature and design of our study, we were unable to determine the specific reasons for zinc testing in our control group. We cannot categorize this group as healthy controls, and we indeed did not mention our control group as such. These are merely patients who were investigated for zinc for reasons other than hair loss. We did exclude patients who had conditions that might interfere with zinc, "such as alopecia areata, cirrhosis, human immunodeficiency virus (HIV), alcohol abuse, inflammatory bowel disease, sickle cell anemia, pregnancy, known zinc deficiency, or hypoalbuminemia 27. Further exclusion criteria included intake of medications that might interfere with zinc levels, such as zinc supplements, oral contraceptives, or oral corticosteroids (taken consistently)". (Materials and methods, p3, 105-109)
We added this to the limitation section in the discussion.
Third, zinc measurements were not performed randomly. In the study group, testing was prompted by hair loss, and in the control group the indications were unknown, but likely related to clinical suspicion of deficiency. This may bias zinc levels downward in both groups. Still, zinc levels in the control group were comparable to those of a healthy reference population.10 (Discussion, p7, 239-243).
Heterogeneity of Hair Loss Diagnoses
The authors pool together multiple etiologies of hair loss (e.g., androgenetic alopecia, telogen effluvium, female pattern hair loss) despite the known differences in their pathophysiology and metabolic profiles. Without subgroup analysis, the findings are overly generalized and potentially misleading. At minimum, results should be stratified by diagnosis.
ANSWER:
Our study's design is a retrospective cohort based on electronic medical records and ICD coding of our patients. Coding was performed by primary care physicians, who might label androgenic alopecia as a general hair loss diagnosis or as telogen effluvium. Since it is a retrospective study and we were unable to ensure proper diagnostic procedures, we decided to pool these diagnoses. It is important to mention that the current literature did not specify a distinct diagnosis among these diagnoses in which zinc deficiency is more established (unlike Alopecia Areata, which was therefore excluded). Following your comment, we have added the following to the conclusion paragraph in the discussion section:
Future large prospective studies on the different etiologies of hair loss and zinc are needed to investigate differences between these diagnoses, and further studies on the efficacy of zinc supplementation in hair loss could elucidate the clinical relevance of zinc in hair loss conditions. (Conclusion, p7, 255-258)
3/ With a sample size of nearly 24,000 patients, statistically significant p-values are expected even for clinically trivial differences. The authors should report effect sizes (e.g., Cohen’s d) and address the magnitude, not merely the presence, of statistical differences. The current interpretation risks a “p-value fallacy.”
ANSWER:
Thank you very much for your comment, we've added the following:
Continuous variables were reported as medians and interquartile ranges (Q1-Q3) in square brackets, and between-group comparisons were performed using the Mann-Whitney test or the Kruskal-Wallis test, with effect sizes reported. (Material and methods, p3, 119-121).
When comparing test results (ferritin, hemoglobin, iron, MCH, TSH, transferrin, transferrin saturation), patients with a hair loss diagnosis had lower ferritin (30.0 [16.8-53.0] vs. 33.0 [16.0-65.0], p-value=0.001) and hemoglobin (12.9 [12.2-13.6] vs. 13.1 [12.3-14.0], p-value<0.001) values. The effect size was small for all the laboratory variables. The comparison of blood test levels of participants with and without hair loss are presented in Table 2. (Results, p4, 151-156)
Effect size was added to table 2 and table 4 – added in the manuscript.
The effect size was small for all the variables we evaluated. The complete comparison is presented in Table 4. (Results, p5, 177-178).
4.Key lifestyle and nutritional factors (diet, supplementation, vegetarianism, alcohol intake) are not addressed, despite being directly linked to zinc levels. This omission limits the strength of any causal inference or even correlation.
ANSWER:
We agree it is important to address these matters, and we believe we have done so, despite the limitations of our study's design. We have excluded every patient whose prior medical history might cause zinc deficiency, including the following alopecia areata, cirrhosis, human immunodeficiency virus (HIV), alcohol abuse, inflammatory bowel disease, sickle cell anemia, pregnancy, known zinc deficiency, or hypoalbuminemia 27. Further exclusion criteria included intake of medications that might interfere with zinc levels, such as zinc supplements, oral contraceptives, or oral corticosteroids (taken consistently). (Materials and Methods, p3, 105-109)
Diet, over-the-counter use of supplements, and vegetarianism – these are all unavailable in the electronic medical record, and we were unable to discuss these matters. This is another limitation of our study, which is retrospective in nature.
We added the following to the limitations section in the discussion:
Fourth, we were unable to investigate the population's nutritional lifestyle, including the possibility of over-the-counter zinc supplementation, as these factors can affect zinc levels, which may be important. (discussion, p7, 243-245).

Round 2
Reviewer 1 Report
Comments and Suggestions for Authors
Accept in present form
Reviewer 2 Report
Comments and Suggestions for Authors
no additional comments or suggestions.
Reviewer 3 Report
Comments and Suggestions for Authors
Thank you for your responses, I approve the revised manuscript.